# Effects of OsAOX1a Deficiency on Mitochondrial Metabolism at Critical Node of Seed Viability in Rice

**DOI:** 10.3390/plants12122284

**Published:** 2023-06-12

**Authors:** Jing Ji, Shuangshuang Lin, Xia Xin, Yang Li, Juanjuan He, Xinyue Xu, Yunxia Zhao, Gefei Su, Xinxiong Lu, Guangkun Yin

**Affiliations:** 1National Crop Genebank, Institute of Crop Science, Chinese Academy of Agricultural Sciences, Beijing 100081, China; jijingmail@163.com (J.J.); shuanglin604@163.com (S.L.); xinxia@caas.cn (X.X.); liyang04@caas.cn (Y.L.); hejuanjuan@caas.cn (J.H.); jorcexu@163.com (X.X.); zhaoyunxia2023@163.com (Y.Z.); sfs923636@gmail.com (G.S.); 2Institute of Agricultural Bioresource, Fujian Academy of Agricultural Sciences, Fuzhou 350003, China; 3College of Biological Science, China Agricultural University, Beijing 100193, China

**Keywords:** alternative oxidase, mitochondria, critical node, rice

## Abstract

Mitochondrial alternative oxidase 1a (AOX1a) plays an extremely important role in the critical node of seed viability during storage. However, the regulatory mechanism is still poorly understood. The aim of this study was to identify the regulatory mechanisms by comparing *OsAOX1a*-RNAi and wild-type (WT) rice seed during artificial aging treatment. Weight gain and time for the seed germination percentage decreased to 50% (*P50*) in *OsAOX1a*-RNAi rice seed, indicating possible impairment in seed development and storability. Compared to WT seeds at 100%, 90%, 80%, and 70% germination, the NADH- and succinate-dependent O_2_ consumption, the activity of mitochondrial malate dehydrogenase, and ATP contents all decreased in the *OsAOX1a*-RNAi seeds, indicating that mitochondrial status in the *OsAOX1a*-RNAi seeds after imbibition was weaker than in the WT seeds. In addition, the reduction in the abundance of Complex I subunits showed that the capacity of the mitochondrial electron transfer chain was significantly inhibited in the *OsAOX1a*-RNAi seeds at the critical node of seed viability. The results indicate that ATP production was impaired in the *OsAOX1a*-RNAi seeds during aging. Therefore, we conclude that mitochondrial metabolism and alternative pathways were severely inhibited in the *OsAOX1a*-RNAi seeds at critical node of viability, which could accelerate the collapse of seed viability. The precise regulatory mechanism of the alternative pathway at the critical node of viability needs to be further analyzed. This finding might provide the basis for developing monitoring and warning indicators when seed viability declines to the critical node during storage.

## 1. Introduction

Mitochondrial redox homeostasis is a key factor affecting seed viability. During the imbibition process of seeds with high viability, the number of mitochondria, as well as the inner membrane and cristae of the mitochondria are significantly improved, and proteins and genes involved in energy and material metabolism, enzyme activity of TCA cycle, electron transport chain and oxidative phosphorylation are also significantly improved [1,2]. However, in the process of seed aging, there are a number of deleterious changes that inhibits the repair of mitochondrial structure and function; the total number of mitochondria (the outer membrane and inner membrane) are difficult to distinguish, and membrane integrity+ decrease the number of matrix, cristae is less than that of high viability seeds and the shape is irregular. The proportion of mitochondria increased and the motility decreased, and the physiological functions, such as electron transport chain, oxidative phosphorylation and marker enzymes, were also lower than normal seeds, and mitochondrial cytochrome *c* was released into the cytoplasm, inducing programmed cell death [3,4,5].

With the decrease in seed viability, the activities of superoxide dismutase, ascorbate peroxidase, monodehydroascorbate reductase and glutathione reductase of mitochondrial antioxidant system have also been shown to decrease significantly. Moreover, the total amount of ascorbic acid and glutathione decrease, and the ratio of oxidized forms also decrease significantly, resulting in the accumulation of reactive oxygen species (ROS) [6,7]. The accumulation of ROS oxidizes lysine, arginine, and proline in the protein side chain, resulting in protein carbonylation modification, inducing protein peptide chain breakage, and the change of the oxidation site to carbonyl [8]. Carbonylation then causes changes in protein structure and loss of function, which are easily degraded by hydrolases, leading to physiological and metabolic disorders of cells and tissues, aging or death [9]. Researchers have shown that when rice seed viability drops to the level of the critical node (CN, 85% germination), ROS accumulate and attack lipids, and 4-HNE and other carbonyl small molecules accumulate, resulting in the production of mitochondrial ATP synthase, malate dehydrogenase, and succinate dehydrogenase. The protein undergoes carbonylation modification, which inhibits the ATP supply required for seed germination [10,11].

Mitochondrial redox homeostasis is closely related to the mitochondrial electron transport chain (mETC). mETC in plants is associated chiefly with the cytochrome pathway and alternate oxidase pathway. In the cytochrome pathway, electrons are transferred from reduced substrates (NADH and succinate) to Complex I or Complex II, and then to ubiquinone, Complex III, cytochrome c (Cyt *c*), Complex IV, and subsequently, the transfer of electrons to O_2_ which generates H_2_O, while protons are pumped from the mitochondrial matrix to the mitochondrial intermembrane space, forming proton kinetic potential on both sides of the inner membrane, thereby driving the synthesis of ATP [12,13]. Under stress, the cytochrome pathway is inhibited, the oxidative capacity of NADH is reduced, the transfer of electrons to ubiquinone is inhibited, the proportion of oxidized ubiquinone is increased, and the transfer of electrons to O_2_ generates O_2_^−^, resulting in oxidative damage, At the same time, it induces the alternative oxidase(AOX) pathway, which facilitates electron transfer to AOX, and the pumping of protons from the mETC membrane arm to generate ATP [14,15,16].

AOX is involved in clearing ROS in the mitochondria [17]. When AOX activity is inhibited, mitochondrial ROS levels increase significantly, whereas overexpression of AOX can enhance the activity of the ROS clearance system [18,19]. AOX can reduce excessive ROS production by regulating electron transfer in the mitochondrial complex. Tobacco plants lacking in AOX were severely inhibited in cell respiration, and showed a significant increase in apoptosis after treatment with H_2_O_2_, salicylic acid, and other agents [20]. In contrast, wild-type tobacco showed a significant increase in AOX content, maintaining a high respiratory rate and inhibiting ROS production [20]. Kühn et al. [21] reported the molecular mechanism by which the redox homeostasis of *aox1a* mutants regulates the cytochrome pathway and the AOX pathway under stress in Arabidopsis.

Different plants contain different numbers of *AOX* genes. For example, the rice genome contains four *AOX* genes, *OsAOX1a*, *OsAOX1c*, *OsAOX1d* (previously named *OsAOX1b*), and *OsAOX1e* [22,23]. Moreover, five *AOXs* (*AtAOX1a*–d and *AtAOX2*) have been identified in Arabidopsis, while *HvAOX1a*, *HvAOX1c*, *HvAOX1d1* and *HvAOX1d2* have been identified in barley [24,25]. *AOX* genes expression are regulated by a number of factors, such as temperature, light, and nutrient availability, among others. *AOX1a* plays an important role in response to stress in both Arabidopsis and rice. *AOX1c* is expressed in all plant developmental stages at a very low level, while *AOX1d* is expressed during the early vegetative stage, flowering, and in senescent leaves [26]. *AOX1e* is not a well-known AOX isoform, and little is known about its specific role in plant metabolism and stress response.

It has been reported that the redox homeostasis of the critical node of rice seed viability was significantly altered, inhibiting the cytochrome pathway, and the expression of *OsAOX1a* was significantly up-regulated compared to gene expression in unaged seeds. Furthermore, the expression of the core subunit of the membrane arm of the mETC complex I was significantly up-regulated, electron and proton transfer ability was improved, and ATP generation was maintained [5,27]. Previous work confirmed that the cytochrome pathway, the accumulation of ROS, and mitochondrial protein carbonylation are the processes that are inhibited during the critical node of rice seed viability, however, AOX1a can be induced to maintain mitochondrial redox homeostasis and enhance the function of ATP generation. Given that the molecular mechanisms underlying rice seed conservation are still unclear, this study will explore the regulatory mechanism of rice seed conservation longevity from the perspective of mitochondrial activity regulation using wild-type and *OsAOX1a*-RNAi rice seeds for artificial aging.

## 2. Results

### 2.1. Comparison of Survival Curves between WT and OsAOX1a-RNAi Seeds after Aging Treatment

We found that the grain length is smaller in seeds with lower levels of *OsAOX1a* expression (Figure 1a). Compared with WT seeds, the seed width, length, and 100-grain weight of *OsAOX1a*-RNAi seeds had decreased by 1.9%, 2.2% and 8.2%, respectively (Figure 1b–d). These results show that the accumulation of dry matter in *OsAOX1a*-RNAi seeds was less than that in the WT seeds.

The initial germination rates of the WT and *OsAOX1a*-RNAi seeds were 98.5% and 96.5%, respectively. Under artificial aging treatment at 40 °C and 75% relative humidity for 30 d, the survival curves of each of the two groups showed the typical inverse S-shaped curve with an initial plateau phase transition state (known as the critical node (CN)), followed by a rapid decline in viability. The shape of the survival curve and most importantly, the time point at which seed survival (germination) declined to 50% (P50), differed between the two groups (Figure 2). The plateau phase of the survival curve was similar between the WT and *OsAOX1a*-RNAi groups. The germination rate of each of the two types of seed fell gradually to 90.0%, 80.0% and 70.0% after 6, 9, and 12 d of artificial aging treatment (Figure 2a), respectively. However, the vigor index of *OsAOX1a*-RNAi seeds during AA treatment was always significantly lower than that of the WT group (Figure 2b). Furthermore, once the critical node was reached (at 12 days of aging), the germination rate of the *OsAOX1a*-RNAi seeds began to decline rapidly compared to the WT seeds. The time required for the seed germination percentage to decrease to 50% (*P50*) was estimated to be 13.6 d for *OsAOX1a*-RNAi and of 14.8 d WT seeds. These results indicated that the storability of *OsAOX1a*-RNAi seeds significantly decreased compared to the WT seeds.

### 2.2. Assessment of Mitochondria Status during Artificial Aging Treatments

To gain deeper insight into the differences in mitochondria status between *OsAOX1a*-RNAi and WT seeds during artificial aging treatments, mitochondria were extracted from seeds that were allowed to imbibe for 48 h at 100%, 90%, 80%, and 70% germination, respectively. The mitochondrial electron transport capacity was evaluated by monitoring NADH and succinate-dependent O_2_ consumption (Table 1). The respiratory rate of mitochondrial significantly decreased with the decrease in seed viability in both seed groups. Compared to the WT seeds at 100%, 90%, 80%, and 70% germination, the NADH-dependent O_2_ consumption of the *OsAOX1a*-RNAi seeds decreased by 62%, 61%, 70% and 53%, respectively; the succinate-dependent O_2_ consumption were decreased by 21%, 28%, 54% and 19%, respectively. In addition, we measured the abundance of AOX1 and Cyt *c*, which plays a crucial role in mitochondrial electron transport (Figure 3). Compared to the WT seeds, the abundance of AOX1 and Cyt *c* in the *OsAOX1a*-RNAi seeds was always lower during the artificial aging treatment. These results suggested that the capacity for electron transport was inhibited in the mitochondria of the *OsAOX1a*-RNAi seeds after imbibition, indicating a decrease in ATP supply. 

In addition, the mitochondrial malate dehydrogenase (MDH) activity and ATP content were measured for both the *OsAOX1a*-RNAi and WT seeds at 100%, 90%, 80% and 70% germination. MDH activity and ATP content were found to decrease significantly with the decrease in seed viability in both the seed groups (Figure 4). However, compared to the WT seeds, the MDH activity of the *OsAOX1a*-RNAi seeds decreased by 14.4%, 14.3%, 13.18% and 24.8%, at 100%, 90%, 80%, and 70% germination, respectively, while ATP content decreased by 47.2%, 60.5%, 53.8% and 51.8%, respectively. These results indicate that the mitochondrial metabolism in the *OsAOX1a*-RNAi seeds after imbibition was weaker than that of the WT group.

### 2.3. The Abundance of Complex I Subunits during Artificial Aging Treatments

Mitochondrial electron transfer chain Complex I is upstream of the AOX pathway and is classified into five functional sections: N module, Q module, P module, CA domain and GLDH domain. The abundance of Complex I subunits was compared between *OsAOX1a*-RNAi and WT seeds at germination rates of 100%, 90%, 80% and 70% (Figure 5).

In the *OsAOX1a*-RNAi seeds, the abundance of 75 kDa and 51 kDa subunits, the core subunits in the N module, were always lower than the corresponding values in the WT seeds during the artificial aging treatment. In addition, the abundance of the 75 kDa subunit in the WT seeds did not significantly change during the aging treatment, however, that in the *OsAOX1a*-RNAi seeds was gradually down-regulated. The abundance of the 51 kDa subunit was down-regulated during treatment, however, the corresponding value in the *OsAOX1a*-RNAi seeds was too low to be resolved. Together, these results indicate that the capacity of electron transfer in *OsAOX1a*-RNAi was inhibited, especially at the time point of critical node of seed viability.

The abundance of the Nad3, Nad4, Nad4L and Nad6 subunits, the core subunits in the P module, showed different patterns between the *OsAOX1a*-RNAi and WT seeds during the aging treatment. The abundance of Nad4 subunit was too low to be resolved in either of the two seed groups. The abundance of Nad3 and Nad6 subunits in the *OsAOX1a*-RNAi seeds was always higher than that in the WT seeds during aging treatment. In addition, the abundance of Nad3 did not significantly change, and the abundance of Nad6 was gradually down-regulated in both the seed groups during the aging treatment. The abundance of the Nad4L subunit in the *OsAOX1a*-RNAi seeds was always lower than that in the WT during the aging treatment. In addition, the abundance of Nad4L subunit in the WT seeds was gradually down-regulated during the aging treatment, but was not significantly changed in the *OsAOX1a*-RNAi seeds.

The abundance of the 39 kDa subunit and the Nad7 subunit, the core subunits in the Q module, was always higher in the *OsAOX1a*-RNAi seeds than in the WT seeds during the aging treatment. The subunits of GLDH and γ-carbonic anhydrase (γ-CA) are the non-conserved subunits of Complex I. The abundance of GLDH subunit was gradually up-regulated in both types of seeds during the treatment. The abundance of the γ-CA subunit in the *OsAOX1a*-RNAi seeds was always lower than that in the WT during the aging treatment.

## 3. Discussion

### 3.1. Low AOX1a Impaired Seed Development and Storability

AOX1a plays an extremely important role in maintaining growth and promoting plant survival under adverse conditions. Many studies have shown that the expression of AOX1a is up-regulated during a number of critical processes in plant growth and development, including seed germination, stress response, flowering and pollination [16,28,29]. The expression level of *AtAOX1a* under suitable conditions had no significant effect on plant vegetative growth, but under low temperature condition, the growth of *AtAOX1a* mutant leaves was shown to be slower and weaker than that of the WT group [30]. Similarly, ripening, respiration, and ethylene production were severely impaired in transgenic *LeAOX*-RNAi tomato plants [31]. In this study, the seed width, length, and 100-grain weight of *OsAOX1a*-RNAi seeds were all less than the corresponding measures in the wild-type seeds (Figure 1). The dry matter accumulation of seeds was found to be positively correlated with the expression level of *OsAOX1a*, and the down-regulation of *OsAOX1a* impaired the regulation of cell life activities and seed development. Although the germination percentage of *OsAOX1a*-RNAi rice seeds after being aged 3, 6 and 9 d did not show a significant difference compared to that of the WT group, the vigor index of *OsAOX1a*-RNAi seeds during the aging treatment was always significantly lower than that of the WT group (Figure 2). In addition, the *P50* of WT is significantly longer than that of *OsAOX1a*-RNAi seeds. These results indicated that the down-regulation of *OsAOX1a* ultimately reduced seed storability. Regulating the expression of AOX1a to adapt to various stresses in the growth environment can help promote plant growth and developmental stability [32,33,34]. However, *OsAOX1a*-RNAi plant is unable to regulate the expression of *OsAOX1a* to cope with various stresses encountered during natural growth conditions. Therefore, it can be concluded that AOX1a is closely involved in the regulation of seed development and storability of the seeds.

### 3.2. Low AOX1a Impaired Mitochondrial Activity during Seed Imbibition

AOX1a is a key component in the mitochondrial alternative chain, whose inactivation could lead to a different response [35,36]. In the present study, we found that in the *OsAOX1a*-RNAi seeds, compared to the WT seeds, the NADH- and succinate-dependent O_2_ consumption were both reduced during the artificial aging treatment, and the expressions of the abundances of AOX1 and Cyt *c* were inhibited (Table 1 and Figure 3). These results are consistent with the fact that the respiratory capacity of AOX1a mitochondria was significantly restricted by the impairment of the alternative-terminal oxidase, the cytochrome oxidase, or both [21,37]. Compared to our previous study, the capacity of O_2_ consumption of the pure mitochondria was higher than the crude mitochondria in the present study, which were isolated from the imbibed rice seed after artificial aging treatment. In addition, the MDH activity and ATP content were reduced in the *OsAOX1a*-RNAi seeds compared to the WT seeds during treatment (Figure 4). These results are consistent with the reported lower activity of the malate-oxaloacetate shuttle in the AOX1a mutant compared to the WT seeds [38]. MDH catalyzes the interconversion of malate and oxaloacetate in mitochondria [39,40]. MDH activity was shown to be reduced in rice seeds during artificial aging treatment [5]. These results indicate that ATP production was severely inhibited in the *OsAOX1a*-RNAi seeds at the critical node of viability. ATP supply was the core factor for seed germination. The insufficient supply of ATP in *OsAOX1a*-RNAi seeds at the critical node of viability might lead to a rapid loss of seed viability. AOX1a was important to maintain mitochondrial ETC and energy performance [41]. When considering the published findings as well as the results of the current study, it appears that mitochondrial metabolism may be inhibited in the *OsAOX1a*-RNAi seeds after imbibition.

### 3.3. Low AOX1a Altered Mitochondrial Electron Transfer Chain Complex I during Seed Imbibition

In our previous study, the alternative pathway of ETC was significantly induced, whereas the cytochrome pathway was inhibited at the critical node of seed viability in rice [5]. Complex I is the entry site for electrons into the ETC through the oxidation of NADH, which is composed of multiple subunits, including three core modules: N (NADH oxidation site) and Q (ubiquinone reduction site) modules, and P (proton transfer site) module [42,43,44]. When Complex I activity is inhibited, the subunits of NADH dehydrogenase II belonging to the P module, and AOX are upregulated under stress conditions, thus forming an alternative pathway for maintaining cellular respiration and energy supply [45,46]. Our previous study had shown that the P module was significantly induced, whereas the N module was inhibited at the critical node of seed viability in rice [27]. In the present study, we also found that the core subunits of the N module were down-regulated, whereas the subunits of the P module were up-regulated in both *OsAOX1a*-RNAi and WT at critical node of seed viability (Figure 5). AOX and the subunits of the N module display dramatic and coordinated responses to stress treatments [47,48]. However, the abundance of Complex I subunits of the WT seeds was higher than that of the *OsAOX1a*-RNAi seeds, which suggests that the capacity of ETC of the WT seed was significantly higher than that of *OsAOX1a*-RNAi seed at the critical node of seed viability; therefore, the capacity of ATP production in the WT seeds was higher than that in the *OsAOX1a*-RNAi seed seeds. This might be the reason for the more rapid loss of viability of the *OsAOX1a*-RNAi seeds compared to the WT after critical node.

## 4. Materials and Methods

### 4.1. Materials and Treatments

The seeds of Japonica rice (*Oryza sativa* L. ssp. *Japonica* cv. “*Nipponbare*”) were used in this study. In order to generate the interfering *OsAOX1a* construct, a 379-bp fragment was amplified from *OsAOX1a* full-length cDNA (AB004864), and then the fragment and its inverted repeat fragment were inserted downstream of a maize *ubiquitin* promoter at the *Bam*HI (NEB), *Sac*I (Takara Bio), *Kpn*I (NEB), and *Spe*I (Promega) restriction sites of the modified pTCK303 vector [49]. Transgenic plants were generated by the constructs were transferred to *Agrobacterium tumefaciens* (strain EHA105), and were transformed into rice callus according to the method of Hiei et al. [50], which resulted to be positive in real-time PCR testing via Hyg^R^ gene. The primers are shown in Table 2. Wild-type and transgenic lines were grown in the research fields located in the Hainan province under natural conditions.

The seeds were sealed in an aluminum foil bag at 40 °C for 7 days to break dormancy. The seed germination rate (%)and the vigor index were determined at 28 °C after 7 d in the dark. Artificial aging treatment was performed at 40 °C and 75% relative humidity for 21 days [5]. The aging curve was characterized by seeds withdrawn from the treatment every 3 days for each of the two genotypes (WT and *OsAOX1a*-RNAi). The time taken for the seed germination rate to decrease to 50% (*P50*) was calculated using the Avrami equation with OriginPro software 2021 [51].

### 4.2. Crude Mitochondria Purification

For mitochondrial purification, rice seeds were allowed to imbibe at 28 °C for 48 h in the dark, after which 1200 embryos were homogenized in grinding buffer (0.3 M mannitol, 2 mM EGTA, 0.5% (*w/v*) polyvinypyrrolidone-40, 0.5% (*w/v*) bovine serum albumin (BSA), 20 mM cysteine, pH 7.5) and the mixture was centrifuged at 2000× *g* for 5 min. The supernatant was removed, and the pellet was resuspended in a wash buffer (0.3 M sucrose, 10 mM TES, pH 7.5) and centrifuged again at 2000× *g* for 5 min. The resulting supernatant was collected and centrifuged at 12,000× *g* for 15 min. The supernatant was then discarded, and the pellet was resuspended in the wash buffer. The mixture was centrifuged at 2000× *g* for 5 min, the supernatant was removed, and the mixture was centrifuged again at 12,000× *g* for 15 min to obtain the mitochondria, from which 4 mg mitochondrial protein could be extracted. Each step of the experiment was performed at 4 °C, and the mitochondria were stored at −80 °C.

### 4.3. Mitochondrial Respiration Rate Assay

Mitochondrial respiration rate was measured using an oxygen electrode (Hansatech, Norwich, UK). Mitochondria (1 mg mitochondrial protein) was added to 1 mL of reaction solution containing 0.3 M sucrose, 10 mM TES-KOH (pH 7.5), 5 mM KH_2_PO_4_, 10 mM NaCl, 2 mM MgSO_4_ and 0.1% (*w/v*) BSA, and was maintained at 20 °C. NADH (10 mM), succinic acid (10 mM) and ADP (0.8 mM) were added according to the procedures reported by Logan et al. [52].

### 4.4. ATP Content Determination

ATP content was determined using ATP Content Assay Kit (Solarbio, Beijing, China) according to the instructions of the manufacturer. Mitochondria pellets were resuspended in 1 mL extraction buffer and centrifuged at 10,000× *g* for 10 min, after which 500 μL chloroform was added to the supernatant, and the mixture was centrifuged at 10,000× *g* for 3 min. The supernatant was collected for ATP detection.

### 4.5. Mitochondrial Malate Dehydrogenase Activity

Mitochondrial malate dehydrogenase activity was based on the reversible reaction of NAD^+^ oxidizing L-malic acid to oxaloacetate, which was determined by monitoring the increase in absorbance at 340 nm and 25 °C, and 0.5 mg of mitochondrial protein was added to the reaction solution (0.1 M potassium phosphate, pH 7.5, 0.2 mM NADH, and 5 mM oxaloacetic acid).

### 4.6. Western Blot t Analysis

The mitochondria protein was extracted from the mitochondrial pellets at 4 °C in the extraction buffer (Tris-HCl, pH 7.5, 2 mM EDTA, 1% (*w/v*) PVP40, and 1 mM DTT), and centrifuged at 20,000× *g* for 15 min. The supernatant was collected and centrifuged at 20,000× *g* for 15 min again. Equal amounts of mitochondrial protein (10 μg per lane) were loaded onto 12% SDS-PAGE gels and transferred to PVDF using the Mini Trans-Blot cell electrophoresis apparatus (Bio-Rad, California, USA). The blotted membrane was blocked for 4 h at room temperature in the Tris-Buffered Saline with Tween-20 buffer (50 mM Tris-HCl, pH 8.2, 0.1% (*v/v*) Tween 20, and 150 mM NaCl) with 2% low-fat milk, and then incubated for 2 h after adding 1:2000 purified protein-specific primary antibody: Nad4, Nad4L, Nad6, and Nad7 (Beijing Protein Innovation, Beijing, China); 51 kDa, 75 kDa, 18 kDa, and 39 kDa (Phytoab, Kansas, USA); and Cyt *c*, AOX1, Nad3, GLDH, and γ-CA (Agrisera, Vännäs, Sweden). A secondary antibody was the anti-rabbit IgG (Agrisera, Vännäs, Sweden) conjugated with horseradish peroxidase diluted 1:5000. LI-COR Odyssey dual-color infrared fluorescence imaging system (LI-COR Biosciences, Nebraska, USA) was used for imaging.

### 4.7. Statistical Analyses

All data were analyzed as a one-variable general linear model procedure (analysis of variance) via SPSS (SPSS Inc., Chicago, IL, USA). Differences between *OsAOX1a*-RNAi transgenic plants and WT controls detected using the Student’s *t*-test at the level of *p* < 0.05, 0.01, and 0.001 were considered as significant, and were labeled as *, **, and ***, respectively.

## 5. Conclusions

In conclusion, we explored the biological mechanisms of rice seed viability at the critical node, with the goal of providing a scientific basis for improved seed conservation. The key finding of this study is that seed development and storability was shown to be impaired in *OsAOX1a*-RNAi seeds. In addition, mitochondrial metabolism and alternative pathways were severely inhibited in *OsAOX1a*-RNAi seeds at the critical node of viability. An insufficient level of ATP was available after imbibition, which led to the accelerated decrease in seed viability, compared to the wild type.

## Figures and Tables

**Figure 1 plants-12-02284-f001:**
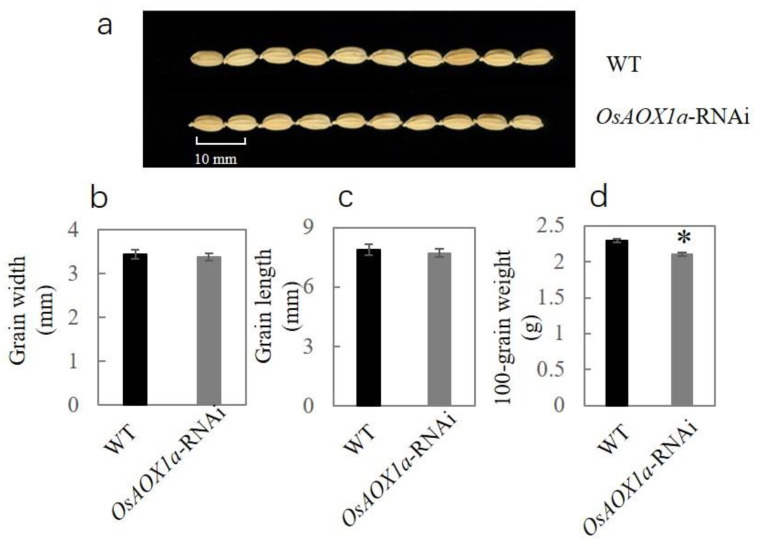
Comparison of wild-type (WT) and *OsAOX1a*-RNAi rice seeds (**a**) with respect to width (**b**) length (**c**) and 100-grain weight (**d**). Data represent the mean ± standard deviation of three independent experiments. Asterisks indicate a significant difference between *OsAOX1a*-RNAi transgenic plants and WT controls via Student’s *t*-test: * *p* < 0.05.

**Figure 2 plants-12-02284-f002:**
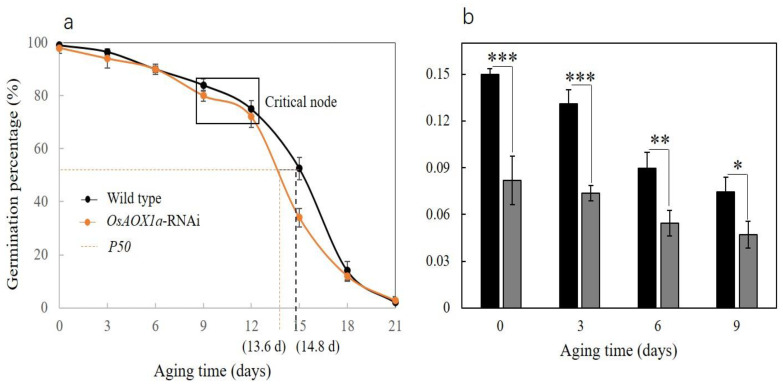
Seed viability curves showing the time for the seed germination percentage decrease to 50% (*P50* (**a**)) and the vigor index (**b**) of wild-type and *OsAOX1a*-RNAi rice following artificial aging treatment. Data represent the mean ± standard deviation of three independent experiments. “……” represents the time when germination percentage decreases to 50%. Asterisks indicate a significant difference between *OsAOX1a* -RNAi transgenic plants and WT controls via Student’s *t*-test: * *p* < 0.05; **, *p* < 0.01; ***, *p* < 0.001.

**Figure 3 plants-12-02284-f003:**
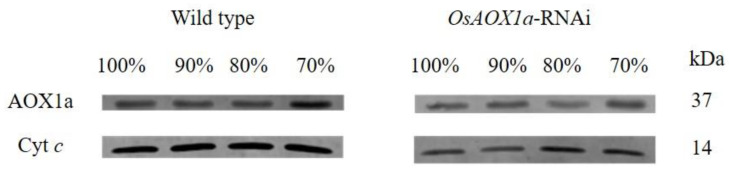
Western blot showing verification abundance of AOX1 and Cyt *c* in mitochondria from wild type and *OsAOX1a*-RNAi seeds at germination percentage of 100%,90%, 80%, and 70% after 48 h imbibition.

**Figure 4 plants-12-02284-f004:**
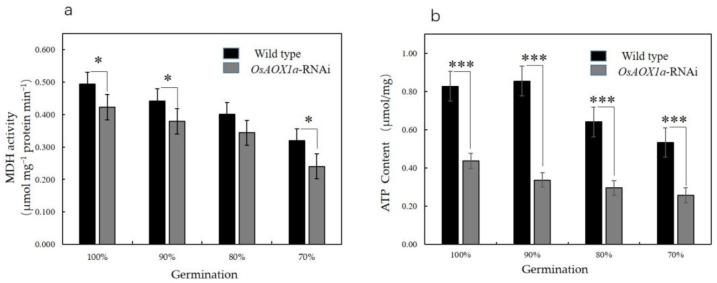
Malate dehydrogenase (MDH) activity (**a**) and ATP content (**b**) in crude mitochondria from wild type and *OsAOX1a*-RNAi seeds at germination percentage of 100%, 90%, 80%, and 70% after 48 h imbibition. Data represent the mean ± standard deviation of three independent experiments. Asterisks indicate a significant difference between *OsAOX1a*-RNAi transgenic plants and WT controls via Student’s *t*-test: * *p* < 0.05; ***, *p* < 0.001.

**Figure 5 plants-12-02284-f005:**
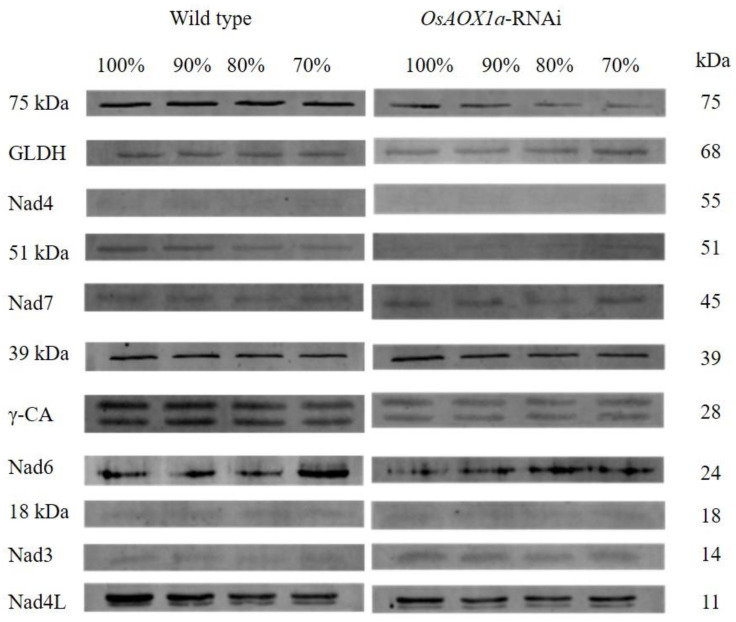
Western blotting showing verification abundance of mitochondrial Complex I subunits in crude mitochondria from wild type and *OsAOX1a*-RNAi seeds at germination *percentage* of 100%, 90%, 80%, and 70% after 48 h imbibition.

**Table 1 plants-12-02284-t001:** Rate of oxygen uptake (nmolO_2·_min^−1^mg protein^−1^) by mitochondria in wild-type and *OsAOX1a*-RNAi seeds at germination percentage of 100%, 90%, 80%, and 70% after 48 h imbibition. O_2_ consumption rates are presented as means ± SD (*n* = 3).

Materials	Substrate	O_2_ Consumption Rate (nmolO_2·_min^−1^mg^−1^ Protein)			
		100%	90%	80%	70%
Wild type	NADH	44.4 ± 3.2	38.1 ± 2.2	18.9 ± 0.6	11.4 ± 2.6
	NADH + ADP	97.8 ± 3.2	65.5 ± 2.9	43.3 ± 3.0	22.7 ± 2.6
	Succinate	24.5 ± 2.0	18.8 ± 0.8	11.5 ± 1.0	7.9 ± 0.7
	Succinate + ADP	40.5 ± 1.9	25.8 ± 1.8	14.2 ± 1.7	9.8 ± 2.6
*OsAOX1a*-RNAi	NADH	17.3 ± 1.9	11.8 ± 1.2	8.3 ± 0.7	6.6 ± 0.4
	NADH + ADP	36.7 ± 0.9	25.2 ± 1.1	12.9 ± 0.7	10.5 ± 1.0
	Succinate	7.9 ± 0.2	6.0 ± 0.3	4.5 ± 0.5	4.1 ± 0.2
	Succinate + ADP	32.0 ± 1.2	18.5 ± 0.7	8.8 ± 0.6	7.9 ± 0.5

**Table 2 plants-12-02284-t002:** Primers used in real-time PCR.

Name	Primer (5′→3′)	Restriction Site
*OsAOX1a-F*	CGGAGCTCGGATCCAACAAAAGAATGGTATGTAG	*Sac*I + *Bam*HI
*OsAOX1a-R*	GGACTAGTGGTACCATATCACTGAGGATGTTTGT	*Spe*I + *Kpn*I
*Hyg-F*	CTATTTCTTTGCCCTCGGAC	
*Hyg-R*	AAGCCTGAACTCACCGCGAC	

## Data Availability

The data presented in this study are available on reasonable request. from the corresponding author.

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
