# Peer review of "Effects of OsAOX1a Deficiency on Mitochondrial Metabolism at Critical Node of Seed Viability in Rice"

_plants, 2023, doi:10.3390/plants12122284_

Round 1
Reviewer 1 Report
The Idea of the manuscript titled with" Physiological regulatory mechanism of AOX1a in Regulating ATP Supply at Critical Node of Viability in Rice" is not new but the work is good. I have some comments on the manuscript as follow;
1-title: should be changed into "Physiological regulatory mechanism of the maternal AOX1a in controlling ATP Supply at Critical Node of Viability in germinated Rice".
2- at the end of the abstract the authors did not mention the importance of their finding for rice germination and its impact on the crop yield.
3-did the authors compared between the maternal AOX1a gene(s) and another genes founded in the chromosomal DNA, the authors should write two lines on the another pathway controlled by chromosomal DNA and could be candidate to be the substitution of the that pathway which regulated by maternal genes to clarify this point for the readers..
4- Seed viability and ATP production in the stored seed should be measured in the embryos of both the transgenic seed and the wild type, and the obtained values should be excised from the germinated seeds in control conditions.
5-I think, studying the activity of AOX1a in the embryonic level will be more conspicuous and the results will be more accurate.
6- The primers DNA sequence references should be listed in the Table 2.
7-The restriction Enzymes and the Promoters, both Origin and company should be listed.
8- In line 279 the authors mentioned that " 1200 embryos were grinded in buffer", which buffer and its contents, please add. Also, the purity and concentration of the obtained mitochondrial DNA should be mentioned and documented.
9- The authors made western blot for the mitochondrial protein and they did not mentioned any methodology how they extracted the mitochondrial protein and how they obtained it in purified form. Please add this.
10- The citation both in discussion and introduction is low and should be increased to enrich the two indicated parts.
11-conclusion; it is copied from the abstract, therefore should be reedit.
Reviewer 2 Report
The research aims at revealing the regulatory mechanisms involved in maintaining mitochondrial redox homeostasis and ATP function of during seed ageing rice seeds. To demonstrate the important role of alternative oxidase 1a at critical node of seed viability, the authors used the original model, involving suppression of AOX1 gene via RNA interference.
Comments and recommendations:
- Please describe the procedure of immunoblot analysis in more details: SDS-electrophoresis, visualisation, the molecular masses of protein bands.
- The phenotypic analysis of seeds should be described in more details (or the corresponding reference should be given).
- Please specify the genotype (name) of rice accessions used in the experiments. The use of only a taxonomic name is clearly not enough in the context of this manuscript.
- Please specify all abbreviation for some terms at the first mention. For example: “SOD, APX, MDHAR and GR” (line 45), “AA treatment” (line 92), P50 (line 15).
- Please verify in the original articles and correct technical errors: replace “OsAOX1a-RNAi Arabidopsis thaliana leaves” by ”AtAOX1a-RNAi Arabidopsis thaliana leaves” (line 205, 206), replace “OsAOX1a-RNAi” by “transgenic 35S-AOX-RNAi tomato plants” (line 207).
- The text requires careful editing and correction of grammatical, stylistic, and technical errors (see lines 72, 79, 93-95, 100-101, 129,130, 152, 153, 168, 181, 242, 304).
Reviewer 3 Report
The manuscript entitled “Physiological regulatory mechanism of AOX1a in Regulating ATP Supply at Critical Node of Viability in Rice” by Ji et al compares the regulatory mechanism of mitochondrial alternative oxidase 1a (AOX1a) between OsAOX1a-RNAi and WT rice seed during artificial aging treatment. This study found that the mitochondrial activities were inhibited after AA treatment in OsAOX1a-RNAi seeds.
This study provides insights into the regulatory mechanism of rice seed conservation longevity from the perspective of mitochondrial activity regulating using WT and OsAOX1a-RNAi rice seeds during aging. Howere, there are many concerns before the manuscript is good for publication.
1. Figure 1 legend: it is not appropriate to show panel number (a, b, c, d) inside the figure. Line 100, there is a redundant word “... with with statistics for…”
Please check all of the rest figure legends.
2.Line 170-174: it is not clear what the authors described. The text must be rephrased to clearly present the results.
1.There are many language errors in the manuscript. The authors must proofread the manuscripts.
Line 98: “.. WT seed”
Line 93-96, Comparing the OsAOX1a-RNAi and wild-type rice seeds harvested in the same, while OsAOX1a low expression seeds have longer grain length smaller (Figure 1a)
Line 90: spelling error, “... during ageing.”
Line 100, Line 145, Line 158, Line254: “OsAOX1a” should be italic font.
Line 112: “comparing to WT seeds”
Line 115: “... were significant decrease …”
Line 131: “In addition to, …”
Line 152 and 153: “... were decrease by 14.4%...”
Line 155: “... weaken than the WT.” the text is incomplete.
Line 170, Line 181, and Line 186: “.... has no significantly changed during AA treatment…”
Line 226: “.. O2 consumption of the pure…” should be O2 consumption.
Line 297: “... for detection ATP.”
I didn’t list all of the errors in the manuscript. Please carefully proofread the whole manuscript.
Reviewer 4 Report
In this manuscript entitled “Physiological regulatory mechanism of AOX1a in Regulating ATP Supply at Critical Node of Viability in Rice” the authors claim that Aox1a regulates the ATP production and function of mitochondria.
The claims are not really supported by the findings, rather it is an extrapolation of the results.
The manuscript is not written well. Many of the sentences are not clear.
Specific comments:
Title: This title is not appropriate. The results of this manuscript don’t really prove the role of aox1 in ATP production.
Abstract: “ In addition to, the reduced abundance of complex I N and P module subunits might showed 21 that the capacity of mitochondrial electron transfer chain was significantly inhibited in the 22 OsAOX1a-RNAi seed at critical node of seed viability. Above results might indicated that the ATP 23 production was impaired in OsAOX1a-RNAi seeds during ageing.”
These is just an assumption or hypothesis. Better to delete these sentences in the abstract. There is no result in this manuscript that prove this.
Line 93: “Compared seed width, length, and 100-grain weight with WT, the OsAOX1a-RNAi seeds 95 decreased by 1.9%, 2.2% and 8.2%, respectively”.
This sentence is not clear. Rephrase it.
Figure: 1: labelling a,b,c,b looks weird. I suggest labelling a, b, c, or d at the left side top of each graph/image.
Line 100: “Comparison of seed width and length (a) between WT and OsAOX1a-RNAi rice, with with 100 statistics for width (b), length (c) and 100-grain weight (d).” This sentence is not correct. Rephrase it.
Line 117: “Seed viability curves and P50……”. Figure legends should be self-supporting. While mentioning first time in each figure, abbreviation should not be used.
Line 119: What statistical analysis was performed?
The manuscript is written poorly. Many of the sentences are not clear.
Round 2
Reviewer 4 Report
Authors have addressed all my comments. I recommend this manuscript for publication.
Moderate editing of English language required